# CX3CL1 Regulation of Gliosis in Neuroinflammatory and Neuroprotective Processes

**DOI:** 10.3390/ijms26030959

**Published:** 2025-01-23

**Authors:** Irene L. Gutiérrez, David Martín-Hernández, Karina S. MacDowell, Borja García-Bueno, Javier R. Caso, Juan C. Leza, José L. M. Madrigal

**Affiliations:** Department of Pharmacology and Toxicology, School of Medicine, Universidad Complutense de Madrid (UCM), Av. Complutense s/n, 28040, Centro de Investigación Biomédica en Red de Salud Mental (CIBERSAM), Instituto de Investigación Neuroquímica (IUINQ-UCM), Instituto de Investigación Sanitaria Hospital 12 de Octubre (Imas12), 28040 Madrid, Spain

**Keywords:** fractalkine, CX3CL1, CX3CR1, neurons, microglia, neuroinflammation, neurodegeneration, chemokines

## Abstract

Among the different chemokines, C-X3-C motif chemokine ligand 1 or CX3CL1, also named fractalkine, is one of the most interesting due to its characteristics, including its unique structure, not shared by any other chemokine, and its ability to function both in a membrane-bound form and in a soluble form, among others. However, undoubtedly, its most relevant characteristic from the neuroscientific point of view is its role as a messenger used by neurons to communicate with microglia. The study of the interaction between both cell types and the key role that CX3CL1 seems to play has facilitated the identification of CX3CL1 as a crucial modulator of microglial activation and a promising target in the fight against neuroinflammation. As a result, numerous studies have contributed to elucidate the involvement of CX3CL1 and its specific receptor CCX3CR1 in the progression of different neuroinflammatory and neurodegenerative processes, with Alzheimer’s and Parkinson’s diseases being the most studied ones. However, the different animal and cellular models used to reproduce the pathological conditions to be analyzed, as well as the difficulties inherent to studies performed on human samples, have hindered the collection of compatible results in many cases. In this review, we summarize some of the most relevant data describing the alterations found for the CX3CL1/CX3CR1 signaling axis in different neurodegenerative conditions in which neuroinflammation is known to play a relevant role.

## 1. Introduction

Chemokines are proteins that function as intercellular messengers. More specifically, the signal transmitted by these messengers stimulates the movement of certain cells towards the areas where chemokines accumulate. It is from this activity that their name originates, as it is derived from the combination of the words chemotaxis, movement in response to a chemical stimulus, and cytokines, proteins that function as messengers between cells. To exert this attraction, after their production by certain cells, chemokines activate specific receptors on other cells, starting a series of processes within these cells that lead to their displacement towards the areas where higher concentrations of chemokines are detected.

Due to the relevance of their chemotactic actions, chemokines have been extensively studied from different perspectives. This has facilitated the discovery of the chemokines documented so far, as well as the advancement in our knowledge of their actions. These actions have been found to include many more than just the propagation of signals controlling the movement of immune cells.

C-X3-C motif chemokine ligand 1, or CX3CL1, also known as fractalkine, is factually unique as it is the only member of the CX3CL group of chemokines, one of the four subfamilies in which chemokines are organized based on the distribution of two of their constituting cysteines. However, in addition to this, other characteristics that are only shared with another couple of chemokines make CX3CL1 stand out from the rest. These include its ability to function both in a soluble form and in a membrane-bound form, as well as its pre-eminent role within the central nervous system [1].

The exclusive production of CL3CL1 by neurons and no other cell type in the brain, together with the location of CX3CR1, its selective receptor, only in microglia, has led to the hypothesis that CX3CL1 is a messenger used unidirectionally by neurons to communicate with microglia [2]. This has been proven to be right and, more interestingly, numerous studies have found that the most intriguing effect of microglial activation of CX3CR1 receptors was the restriction of microglial activation and subsequent neuronal damage [3]. Therefore, this communication system became a promising target whose modulation could have a profound impact on the progression of all pathologies in which neuroinflammation is involved to a greater or lesser degree.

This phenomenon led to the proliferation of studies focused on the analysis of the mechanisms through which CX3CL1 exerts its actions in the central nervous system. This review is a compilation of the most relevant information generated by these studies and aims at facilitating the knowledge of CX3CL1 characteristics and its actions in different neurodegenerative conditions.

## 2. CX3CL1 Is Unique Among Chemokines

Unlike other more promiscuous chemokine receptors, CX3CR1 only recognizes CX3CL1 as its ligand [4]. This fact facilitates the study of CX3CL1 actions due to the relative ease to suppress CX3CL1-receptor-mediated actions using, for example, CX3CR1-KO mice or a specific CX3CR1 inhibitor.

Microglia are considered by many authors as the sole cell type expressing CX3CR1 within the CNS [5]. In fact, CX3CR1 is commonly accepted as a microglial marker. Because of this, CX3CR1-GFP mice, in which the CX3CR1 gene is replaced by a green fluorescent protein (GFP) reporter gene, are used both as a way to easily label and detect microglia through fluorescence and to eliminate CX3CR1-mediated actions [6]. However, after the initial discovery of CX3CR1 expression on rat hippocampal neurons [7] and its role in neuronal protection [8] by the same research group, additional studies have confirmed that CX3CR1 is also expressed by certain neurons in the hippocampus and in the striatum. In these cells, CX3CR1 expression has been associated with a larger degree of neuronal apoptosis in models of permanent middle cerebral artery occlusion [9]. In addition, it has been proposed that an altered expression of neuronal CX3CR1 could mediate motor function deficits in mice [10].

In addition, CX3CR1 has also been detected in rat [11,12] and human [13] astrocytes. The activation of this receptor on astrocytes stimulates their production of some unknown factor or factors that induce microglial proliferation, while astrocytes do not alter their proliferation rate after CX3CL1 treatment [14].

Therefore, it does not seem correct to use CX3CR1 as a single microglia marker, nor can it be accepted that the blockade of CX3CR1 by genetic alterations or other means only affects the microglia cells within the CNS. However, aside from its chemoattracting properties, the most relevant effects of CX3CL1, from a neuroinflammation point of view, seem to be those mediated by the activation of microglial CX3CR1. In fact, only a couple of years after CX3CL1 was discovered [15], its power to reduce the microglial secretion of proinflammatory mediators was demonstrated [16].

After this, due to the involvement of neuroinflammation in numerous neurodegenerative processes, the potential exploitation of CX3CL1’s anti-inflammatory actions became a very interesting research field for those seeking out new therapeutic strategies applicable to different neurological diseases.

## 3. CX3CL1 Production in Alzheimer’s Disease

Kim et al. were among the first to explore the involvement of CX3CL1 in Alzheimer’s disease. For this purpose, they quantified the concentration of CX3CL1 in plasma from healthy subjects and patients with mild cognitive impairment or different degrees of Alzheimer’s disease. This allowed them to detect an increased concentration of CX3CL1 in the plasma that paralleled the progression of the disease, but only to a certain point, because the levels of CX3CL1 in the most advanced cases of Alzheimer’s disease analyzed, while being higher than those observed in the healthy subjects, were lower than those found in the intermediate stages of the disease [17]. However, an approach performed using Tg2576 mice, which overexpress a mutant form of the amyloid precursor protein (APP) with the Swedish mutation, found reduced levels of CX3CL1 in the cortex and hippocampus of these mice at 9 months of age. Interestingly, this reduction disappeared as the mice aged, and no differences could be detected at 11 or 17 months of age [18]. While many factors could contribute to the differences between both studies, it is worth noting that the progression of the damage inverted the tendency of CX3CL1 modifications in both cases. This has been proposed to be due to the loss of neurons, which constitute the main source of CX3CL1 in the central nervous system.

Later analyses of expression confirmed the elevation of CX3CL1 levels in the hippocampus of Alzheimer’s disease patients [19]. More interestingly, in agreement with the variations found in plasma levels, also on this occasion, CX3CL1 was found to be more abundant in the samples obtained from intermediate Alzheimer’s disease cases, while it was reduced in the advanced ones [20]. This indicates that a correlation between CX3CL1 production in the brain and its presence in the blood may exist. Furthermore, when peripheral blood mononuclear cells (PBMCs) obtained from human donors were incubated with a blood–brain barrier in vitro model, those cells obtained from Alzheimer’s disease patients stimulated the production of CX3CL1 in both sides of the barrier [21]. Once again, the effect of PBMCs was larger when these cells came from patients with mild Alzheimer’s disease than when they were obtained from advanced cases.

Considering these reports, and the apparent correlation between brain and blood levels, it has been proposed that plasma concentrations of CX3CL1 could be used as an additional marker to diagnose Alzheimer’s disease.

Cerebrospinal fluid constitutes an alternative to blood samples that may report the actual status of the brain more accurately than plasma. Rat studies demonstrate that amyloid β (Aβ) injection in cerebral ventricles leads to the accumulation of microvesicles containing CX3CL1 in the cerebrospinal fluid [22]. However, contradictory results have been described when the concentration of CX3CL1 was measured in cerebrospinal fluid samples obtained from patients suffering from mild cognitive impairment or Alzheimer’s disease. While the earliest study found a reduction in CX3CL1 concentration in mild cognitive impairment or Alzheimer’s disease patients [23], more recent studies have described the existence of higher concentrations of CX3CL1 in cerebrospinal fluid and blood from Alzheimer’s disease and mild cognitive impairment patients [24,25]. Despite the disparity regarding the main conclusion of both articles, it is worth noting that, in all cases, the accumulation of CX3CL1 was lower in the most advanced Alzheimer’s disease cases analyzed. This fact, in addition to supporting the apparent decline of CX3CL1 accumulation paralleling the progression of the disease described above, may highlight the relevance of carefully considering the degree of neurodegeneration in these analyses, because the lack of sufficient intermediate cases or the inaccurate assessment of the degree of degeneration in some of them may prevent the observation of the trend that seems to exist for the production of CX3CL1 in Alzheimer’s disease patients (see summary in Table 1).

## 4. CX3CR1 Production in Alzheimer’s Disease

Since the modifications in the concentration of either the ligand or the receptor in many ligand–receptor systems are attempted to be compensated by an alteration in the production of the other component, it is interesting to investigate if this balancing system also exists for CX3CL1 and CX3CR1. This was analyzed in one study in which the authors injected Aβ fibrils directly into the rat hippocampus. In this way, they found that the blockade of CX3CR1 signaling in the rat hippocampus with small interfering RNA inhibits Aβ-induced upregulation of CX3CR1 [26], suggesting that the loss of CX3CL1 is detected by the microglia as a signal to lower the expression of its receptor. However, this study also demonstrated that the presence of Aβ induces the expression and synthesis of CX3CR1 in the rat hippocampus. Therefore, considering the above-mentioned data, Alzheimer’s disease could be associated not only with an increase in the CX3CL1 ligand, but also in its receptor, supporting the hypothesis of a parallel modification of CX3CL1 and CX3CR1. Our analysis allowed us to confirm that the induction of CX3CR1 also takes place in the 5xFAD mouse model characterized by its elevated expression of the amyloid precursor protein (APP). However, more relevantly, we were also able to detect this induction in human brain samples obtained from Alzheimer’s disease patients [27]. The measurements performed on 3xTg-AD mice also revealed an increase in CX3CR1 expression in their brain cortices at 9 months of age, although the opposite was detected in 3-month-old mice [28]. Interestingly, a reduction in CX3CR1 expression has been described in blood samples from human Alzheimer’s disease patients [29], indicating that inverse changes between the brain and the periphery may occur in Alzheimer’s disease and supporting the potential of CX3CR1 measurements as an alternative diagnostic tool.

## 5. Effects of Altered CX3CL1 Production in Alzheimer’s Disease Models

Given the signaling nature of CX3CL1 and its altered levels in Alzheimer’s disease, this chemokine has been considered as a potential tool that may alter the progression of the pathology. For this reason, different strategies have been used to increase or reduce the synthesis of CX3CL1 and explore if it has an effect on Alzheimer’s disease models.

The overexpression of CX3CL1 using adeno-associated viral vectors reduced microglial activation, neurodegeneration, and the deposition of tau in rTg4510 mice, a model characterized by the expression of high levels of tau. However, this treatment did not seem to have an effect on APP/PS1 mice, a model based on the accumulation of Aβ [30]. This research group further analyzed the therapeutic potential of CX3CL1 using viral vectors to infect cells lining the ventricular system. Using the cerebrospinal fluid to distribute CX3CL1 through the CNS, the authors found that the elevation of CX3CL1 levels improved cognitive functioning in the same tauopathy model [31].

Interestingly, in a related study that used the APP/PS1 mouse model but an opposite strategy, such as the use of CX3CL1-deficient mice, the authors could observe a reduced deposition of Aβ in the cortex and hippocampus of the mice analyzed. However, contrary to this apparently beneficial effect on Aβ, CX3CL1 suppression also resulted in an elevation of phosphorylated tau protein levels [32]. Furthermore, this study also compared the differences between mice in which only membrane-anchored CX3CL1 was suppressed and other mice in which both soluble and membrane-anchored CX3CL1 forms were removed. This comparison demonstrated that the reduction in Aβ and the increase in phosphorylated tau could only be attributed to the membrane-anchored form of CX3CL1.

A later study analyzed mice in which only the chemokine domain was expressed while suppressing the mucin stalk of CX3CL1. This mere alteration reduced the expression of the microglial CX3CR1 receptor, and it also stimulated the production of tau protein and the activation of microglia when exposed to LPS. However, when the accumulation of phosphorylated tau was analyzed after these mice were crossed with hTau mice (expressing only human tau isoforms), it could be observed that the production of only the chemokine domain from CX3CL1 did not result in significant differences compared to the complete suppression of CX3CL1 [33].

CX3CL1 can be cleaved by different proteases, including ADAM-10, ADAM-17, cathepsin S, and BACE1. The action of BACE1 generates a membrane-anchored C-terminal fragment, which can be further cleaved by γ-secretase, releasing the intercellular domain. This domain has been shown to reduce the loss of neurons and the accumulation of Aβ in the 5xFAD mouse model of Alzheimer’s disease [34]. This seems to be signaled through the translocation of the intercellular fragment to the cell nucleus and its action as a regulatory factor that stimulates the expression of certain genes, leading to increased neurogenesis, such as TGF-β2 and TGF-β3, and the subsequent activation of their downstream signaling molecule Smad2 [35]. In addition, the CX3CL1 intracellular domain has also been proposed to exert its neuroprotective actions against Aβ through the activation of insulin-like growth factor receptors [36].

Based on the results provided by these studies, it can be concluded that CX3CL1 plays a significant role in the progression of Alzheimer’s disease. Also, considering the relevance of the balance of certain factors present in the brain and their potential induction of Alzheimer’s disease [37], the preservation of the mechanisms responsible for the maintenance of adequate levels of this chemokine seems to be a relevant goal for treatments aimed at preventing the development of neuroinflammation and the damage associated with it.

## 6. Effects of Altered CX3CR1 Production in Alzheimer’s Disease Models

Due to the almost exclusive production of CX3CR1 by microglia being within the central nervous system, as mentioned above, this receptor is commonly accepted as a specific microglia marker. This fact, combined with the accessibility to CX3CR1-knockout mice, has facilitated the completion of numerous studies focused on the analysis of CX3CR1 functions in healthy brains and in different models of neurological pathologies. The most commonly used CX3CR1-KO mice are those initially developed by Littman’s research group through the replacement of the CX3CR1 gene by a green fluorescent protein reporter gene [6]. This alteration, apart from suppressing CX3CR1 synthesis, and thanks to the presence of a fluorescent label, also facilitates the identification of CX3CR1-expressing cells, with microglia being the most abundant ones in the brain.

One of the first studies on the role of CX3CR1 in Alzheimer’s disease used CX3CR1-KO mice that were crossed with a triple-transgenic mouse model carrying presenilin, APP, and tau mutations. The resulting animals suffered less neuronal loss than those in which CX3CR1 production was not altered [38]. This result was somewhat surprising considering the preventive actions of CX3CR1 activation on microglial activation and the deleterious effects of gliosis. Nevertheless, later studies confirmed the neuroprotective effects of CX3CR1 suppression in different Alzheimer’s disease mouse models [39,40]. These effects included a reduction in both Aβ deposition and the number of microglia surrounding Aβ plaques, while IL1β mRNA levels were elevated. In support of this, when CX3CR1 was removed from CRND8 mice, which overexpress human APP, contrary to previous findings, the number of microglia cells surrounding the amyloid plaques increased, but a reduction in amyloid deposits similar to previous published findings was observed [41]. In this case, it was also noticed that CX3CR1-lacking microglia were not able to phagocyte fibrillar Aβ, while they were very efficiently uptaking protofibrillar forms of this protein. However, a study performed on APP^swe^/PSEN1^dE9^ mice could not observe any notable differences between WT and CX3CR1 haplodeficient mice [42].

CX3CR1 deficiency has also been shown to impair the uptake and degradation of tau protein by microglia [43]. This is probably related to the data obtained from the periphery, since CX3CR1 expression is reduced in monocytes and natural killer cells obtained from human carriers of pathogenic variants of tau [44].

It is known that tau can also bind directly to CX3CR1 and promote its internalization in microglia in this way [45], thus increasing the elimination rate of extracellular tau, but most likely competing with CX3CL1 and reducing, therefore, the signaling exerted by this chemokine.

Since the intraneuronal deposits of hyperphosphorylated tau proteins constitute one of the two main features of Alzheimer’s disease in combination with Aβ, the apparent induction of tau and Aβ phagocytosis by CX3CR1 activation could be a protective mechanism as relevant as the well-known limitation of microglial activation.

As mentioned above, in addition to its well-known presence in microglia, CX3CR1 has also been detected in hippocampal neurons, although in lower amounts [8]. This has allowed us to analyze the specific contribution of neuronal CX3CR1 to the neuronal damage elicited by Aβ. We also observed that, in isolated neurons, CX3CR1 suppression renders these cells more resistant to the cytotoxic actions of Aβ [46]. Therefore, the apparent harmful effects of CX3CR1 activation occurring in situations resembling Alzheimer’s disease may not be attributable exclusively to microglial activity, and instead they could be due to a combination of the actions of different cells present in the brain.

While those studies focused on the measurement of Aβ accumulation, and microglial proliferation seemed to confirm the beneficial effects of CX3CR1 removal in Alzheimer’s disease mouse models, other studies have demonstrated that this alteration could potentiate the behavioral deficits that characterize this disease, with memory impairment being the most relevant [47]. These results are not completely opposed to those focused on the quantification of Aβ and neuroinflammation, but they are more expectable, considering that the suppression of a receptor, which has apparently evolved to mainly facilitate the communication between neurons and microglia, has negative consequences. In fact, in the absence of other genetic alterations or external injuries, mice lacking CX3CR1 exhibit diverse behavioral deficits, including those impacting memory, learning, and motor skills [48], that can be reversed through the blockade of IL1β, supporting the idea of CX3CR1’s duty as a restrictor of microglial activation.

This protective effect of the IL1β blockade has previously been observed through in vitro and in vivo analyses that demonstrated how the absence of CX3CR1 results in increased hyperphosphorylation and aggregation of tau protein, with IL1β activation being required for this [49].

The damage caused by tau to neurons stimulates their expression of CX3CL1 [19], once again supporting the role of this chemokine as a signal generated from injured neurons in order to prevent further damage that could result from microglia responding to this alteration. Indeed, in this publication, the degree of microgliosis and astrogliosis found to be caused by tau administration was larger in the CX3CR1-knockout mice.

Furthermore, more recent analyses have concluded that the suppression of CX3CR1 in 5xFAD mice increases the deposition of Aβ, causes neuritic dystrophy, and exacerbates tau pathology, neuronal loss, and cognitive impairment [50]. The more recent publication date of this study and its use of more modern and refined techniques contribute to increasing its reliability, in comparison to some opposing studies, especially considering that one of those studies presenting opposing conclusions [40] was performed by the same research group.

Altogether, these data confirmed that the lack of CX3CR1 definitively alters microglia behavior; however, the disparity of conclusions also underlined the need for additional research in order to advance in the elucidation of the extent of CX3CL1/CX3CR1regulatory actions in the brain.

## 7. CX3CL1 in Parkinson’s Disease

Among the different features that Parkinson’s and Alzheimer’s diseases have in common, neuroinflammation is one of the most interesting from a pharmacological point of view. For this reason, similarly to Alzheimer’s disease, the analysis of the CX3CL1/CX3CR1signaling system in Parkinson’s disease constitutes a relevant area of study. One of the earliest works to focus on this used the intraperitoneal injection of 1-methyl-4-phenyl-1,2,3,6-tetrahydropyridine (MPTP) in mice as a Parkinson’s disease model. Here, the authors observed that the MPTP-induced neuronal loss was greater in CX3CR1 mice and in CX3CL1-deficient mice [51]. According to this, the responses mediated by CX3CL1 and CX3CR1 would prevent the progression of the damage associated with Parkinson’s disease within the central nervous system. Therefore, an activation of this communication system could be expected as an initial response to the detection of the alterations characteristic of this disease. In fact, in a later study in which 1-methyl-4-phenylpyridinium (MPP) was injected into the substantia nigra of rats, an induction of both CX3CL1 and CX3CR1 synthesis was detected in the neurons and microglia, respectively [52]. Interestingly, in this case, the administration of exogenous CX3CL1 directly into the substantia nigra caused microglial activation, dopaminergic cell depletion, and motor dysfunction, leading the authors to conclude that CX3CL1 could be responsible for some of the deleterious alterations commonly observed in Parkinson’s disease cases.

Most of the subsequent studies demonstrate a protective role for CX3CL1 and CX3CR1 actions. In these, it was described that the administration of exogenous CX3CL1 into the striatum reduced the microglial activation and loss of neurons in a Parkinson’s disease rat model based on intrastriatal administration of 6-hydroxydopamine [53].

The use of adeno-associated, virus-mediated gene therapy was also used to express CX3CL1 in CX3CL1-knockout mice. In this way, it was demonstrated that the soluble form of CX3CL1 reduces motor alterations, as well as dopaminergic neuronal loss and neuroinflammation in the 1-methyl-4-phenyl-1,2,3,6-tetrahydropyridine (MPTP) Parkinson’s disease model [54]. This research group further confirmed the neuroprotective actions of soluble CX3CL1 using an adeno-associated, virus-mediated synuclein model of Parkinson’s disease [55]. However, the use of this synucleinopathy model by a different group allowed them to observe that the deletion of CR3CR1 reduced the inflammatory response, with this probably being due to a reduced uptake of α-synuclein by microglia lacking CX3CR1 [56].

An alternative model of Parkinson’s disease was developed through the overexpression of A53T, a mutated form of α-synuclein in mice. The combination of this model with the genetic suppression of CX3CR1 resulted in an exacerbation of the neurodegeneration characteristic of this disease [57].

A review of the different publications focused on the role of this chemokine in Parkinson’s disease (Table 2) suggests that CX3CL1 actions contribute to preventing the loss of neurons and the excessive activation of microglia that may also result in additional brain damage. However, the opposing data cannot be ignored. While this discrepancy could be related to the use of different methodologies, it could also be due to the complexity of the alterations that take place during the usual evolution of the disease. Indeed, CX3CL1 levels in serum samples obtained from Parkinson’s disease patients were found to be elevated during the initial stages of the disease. However, as the disease progressed, CX3CL1 levels were reduced [58]. Interestingly, the opposite fluctuation has been observed for CX3CR1 in a mouse model based on the injection into the substantia nigra of recombinant adeno-associated virus encoding human α-synuclein, with CX3CR1 expression being reduced during the early phases after the treatment and elevated in the late ones [59].

## 8. CX3CL1/CX3CR1 Polymorphisms and Risk for Other Neurological Diseases

The CX3CR1 human polymorphism I249/M280 is present in approximately 20% of the population. The protein produced by this polymorphism exhibits a reduced affinity for CX3CL1, which leads to diminished regulatory effects on microglia. Consequently, the presence of a hypofunctional variant of CX3CR1 has been associated with the risk of developing different neurodegenerative diseases.

Amyotrophic lateral sclerosis patients who carry one or two copies of the CX3CR1-Val249Ile allele experience a more rapid disease progression and shorter survival than those with wild-type Cx3cr1 [60]. In multiple sclerosis patients, CX3CR1-Val249Ile polymorphisms have revealed that the *Cx3cr1 Ile249 Thr280* haplotype could have a protective effect by impairing the switch of multiple sclerosis from the relapsing–remitting type into the secondary progressive type [61,62], and patients with the variant in both alleles (homozygosity) have a higher risk for disability.

Individuals who carry the CX3CR1-Thr280Met polymorphism, which is linked to a deficit in cell migration, present an increased risk of developing age-dependent macular degeneration [63,64,65]. However, the association of the CX3CR1-Val249Ile variant with this disease remains controversial.

Apart from the roles of the CX3CL1/CX3CR1 axis in the diseases mentioned above, this chemokine and receptor system seem to be implicated in the pathogenesis of other brain diseases, such as diabetic retinopathy, Huntington’s disease, ischemia, neuropathic pain, traumatic brain injury, and epilepsy.

In a murine model of amyotrophic lateral sclerosis, the absence of CX3CR1 accelerates the progression of the disease, leading to rapid and increased neuronal cell death [51,66,67]. In this model, the lack of CX3CR1 enhances the activation of nuclear factor kappa B (NFκB) and impairs both the autophagy–lysosome degradation pathway and autophagosome maturation, resulting in significant damage to motoneurons.

The CX3CL1/CX3CR1 axis also plays a pivotal role in regulating microglial clearance of myelin debris and influencing demyelination/remyelination processes. In a cuprizone-induced demyelination model of multiple sclerosis, CX3CR1-deficient mice exhibit impaired microglial migration and phagocytosis, resulting in persistent myelin debris and defective remyelination with aberrant myelin patterns [68]. Similarly, transgenic mice express a CX3CR1-loss-of-function variant or CX3CL1-deficient mice demonstrate exacerbated demyelination and significantly delayed or incomplete remyelination compared to wild-type mice [69]. In a rat experimental autoimmune encephalomyelitis model, elevated CX3CL1 and CX3CR1 levels in the dorsal root ganglia and the spinal cord were associated with neuropathic pain and neurological impairment [70].

Studies of CX3CR1-KO and CX3CL1-KO mice reveal that a disrupted CX3CR1/CX3CL1 signaling pathway exacerbates diabetic retinopathy damage through microglial activation, neuronal cell loss, and vascular impairment. The use of the recombinant adeno-associated virus expressing the soluble form of CX3CL1 provided neurovascular protection, reduced inflammation, and preserved optic nerve health by regulating microglia-mediated inflammation [71]. This therapeutic approach also prevented fibrinogen leakage, a hallmark of vascular damage, and improved visual acuity at both 4 and 10 weeks of diabetes induction [72]. Additionally, the positive effects of microglia depletion in models of diabetic retinopathy appear to depend on CX3CR1. In both CX3CR1-knockout mice and mice with a loss-of-function variant of the human CX3CR1 gene, the depletion of microglia did not reduce retinal degeneration or alter the morphology of microglia, as observed in wild-type CX3CR1 mice [73].

These data consistently highlight the role of the CX3CL1/CX3CR1 axis in modulating microglial activity, promoting debris clearance, and facilitating tissue repair processes, such as remyelination. In this way, novel therapeutic strategies focused on this chemokine, such as the administration of soluble CX3CL1 via the recombinant adeno-associated virus, have demonstrated promising neuroprotective effects.

The elevation of glutamate levels resulting in excitotoxicity is a common component in the neurodegenerative disorders reviewed above and many other pathological conditions affecting the central nervous system. The involvement of CX3CL1 in this process was demonstrated using neuronal cultures that were exposed to toxic concentrations of glutamate. This treatment caused a cleavage of CX3CL1 from the neuronal membranes that was detectable two hours later and led to the complete loss of membrane-associated CX3CL1 twenty-four hours after the excitotoxic treatment [74]. The release of CX3CL1 by neurons as a response to glutamate agrees with the role of this chemokine as a damage signal generated by endangered neurons, but it also has been demonstrated to directly act on neurons, reducing the degree of apoptosis caused by glutamate. This protection has been demonstrated to be mediated by the reduction in NMDA-induced calcium influx [75]. Later studies showed a similar protection when neurons were treated with glutamate [76] and advanced in the elucidation of the mechanisms implicated, demonstrating that the activation of the ERK12 and PI3K/Akt pathways by CX3CL1 are required to observe this neuroprotection against excitotoxic insults [75,76]. In addition, the activation of adenosine receptor 1 has also been proposed as a requirement for CX3CL1 neuroprotective activity against glutamate [77,78].

In addition to its effects on neurons, CX3CL1 actions on astrocytes have also been observed to be required for its protective actions against glutamate in neurons. This seems to be due to an increase in the activity of astrocytic glutamate transporter 1 (GLT-1) induced by CX3CL1, but also requires the presence and activity of adenosine receptor 1 [79].

As mentioned above, while CX3CR1 is mainly produced by microglia, it can also be found in certain neurons. Based on this, the contribution of this receptor to the neuronal death caused by excitotoxic insults was analyzed in CX3CR1-deficient mice. These analyses indicate that the suppression of CX3CR1 reduces the degree of neuronal death caused by middle cerebral artery occlusion, a commonly used model of ischemic stroke. Additionally, in vitro studies performed on hippocampal neurons obtained from CX3CR1-deficient mice also indicate that the lack of CX3CR1 protects these neurons from excitotoxicity caused by direct treatment with glutamate [9,80]. According to the authors’ conclusions, while some of this protection could be due to the reduced recruitment of peripheral macrophages within the brain parenchyma, it cannot be applied to in vitro analyses where neurons were isolated.

An alternative model of excitotoxicity consisting of intrastriatal injection of kainic acid was also used to test the potential role of CX3CR1 in neuronal death caused by excitotoxicity. In this way, an exacerbated loss of neurons could be observed in CX3CR1-deficient mice [81].

Therefore, it is interesting to observe how CX3CL1 acting on neurons can protect them from excitotoxic insults, but the removal of the specific receptor CX3CR1 could have different effects, opening new routes for the exploration for CX3CL1’s and CX3CR1’s potentially independent roles on neurons.

## 9. Conclusions

Among the numerous agents involved in the generation and progression of neuroinflammatory processes, chemokines constitute a group of the most important mediators, mainly due to their ability to attract different types of cells and facilitate, in this way, the interaction of these cells with the source of alteration. Due to this key role and to the relevance that neuroinflammation has been found to have in different pathologies causing neurodegeneration, chemokines have become the focus of numerous studies searching for new means to curb the excessive activation of glial cells and the deleterious consequences associated with it.

Among the different chemokines, CX3CL1 stands out due to certain characteristics that render it unique. However, CX3CL1’s most interesting feature for those searching for microglial modulators is the direct connection it seems to have with microglia, as these cells are the main producers of its only receptor, CX3CR1.

The use of different strategies that increase or reduce the activity of CX3CL1 or CX3CR1 in animal models of certain neurodegenerative pathologies, combined with in vitro analyses, have facilitated the advancement in the knowledge of the different mechanisms through which this chemokine and its receptor can alter the progression of neuroinflammation and the neuronal loss associated with it. In addition, the use of human samples has demonstrated that alterations in CX3CL1 and CX3CR1 expression and synthesis are quantifiable indicators of the progression of certain neurological diseases. Therefore, the data available so far may serve as the basis for new therapeutical strategies based on the modulation of the CX3CL1/CX3CR1 communication system.

Unfortunately, several contradictory results regarding the protective or harmful effects of this chemokine have been published, making it difficult to reach conclusions with an acceptable degree of certainty. For this reason, in order to facilitate the advancement of this research line, it is still necessary to increase the number of studies that repeat certain experimental conditions, as this would facilitate their comparison. While this seems complicated, due to the particular curiosity of each investigator, the potential interest of pharmaceutical companies looking for new strategies to reduce neuroinflammation could lead to the type of uniform and therapeutically focused studies needed to generate enough data to support the preliminary conclusions available so far and hopefully result in new therapies for neurodegenerative diseases.

## Figures and Tables

**Table 1 ijms-26-00959-t001:** CX3CL1 alterations described in Alzheimer’s disease.

Reference	Model/Sample Type	Results
Kim, 2008 [17]	Human plasma	Increased CX3CL1 in AD patients
Duan, 2008 [18]	Tg2576 mice	Decreased CX3CL1 in cortex and HC
Lastres, 2014 [19]	Human HC	Increased CX3CL1 in AD patients
Strobel, 2015 [20]	Human HC	Increased CX3CL1 in AD patients
Karimi-Zandi, 2022 [22]	Aβ ICV injection in rats	Increased CX3CL1 microvesicles in HC
Perea, 2018 [23]	Human CSF	Decreased CX3CL1 in AD patients
Kulczynska-Przybik, 2020 [24]	Human CSF and blood	Increased CX3CL1 in AD patients
Bivona, 2022 [25]	Human CSF	Increased CX3CL1 in AD patients
Wu, 2013 [26]	Aβ injection in rat HC	Increased CX3CR1 expression
Gonzalez-Prieto, 2021 [27]	Human brain cortex	Increased CX3CR1 in AD patients
Fernandes 2022, [28]	3xTg-AD mice	Increased CX3CR1 in AD model
Kim, 2022 [29]	Human blood	Decreased CX3CR1 in AD patients

HC: hippocampus, ICV: intracerebroventricular, CSF: cerebrospinal fluid, AD: Alzheimer’s disease.

**Table 2 ijms-26-00959-t002:** CX3CL1 and CX3CR1 alterations and effects described in Parkinson’s disease.

Reference	Model/Sample Type	Results
Cardona, 2006 [51]	MPTP mice	Increased neurotoxicity in CX3CR1-KO mice
Shan, 2011 [52]	MPP rats	Induction of CX3CL1 and CX3CR1
Pabon, 2011 [53]	6-OHDA in rats	Neuroprotecion by CX3CL1 in striatum
Morganti, 2012 [54]	MPTP mice	Neuroprotecion by CX3CL1 in SN
Nash, 2015 [55]	Synucleinopathy rats	Neuroprotecion by CX3CL1 in SN
Thome, 2015 [56]	Synucleinopathy mice	CX3CR1 supression reduces neuroinflammation
Castro-Sanchez, 2018 [57]	A53T overexpression mice	CX3CR1 supression increases neurodegeneration
Gupta, 2022 [58]	Human serum	Increased CX3CL1 in PD patients

MPTP: 1-methyl-4-phenyl-1,2,3,6-tetrahydropyridine, MPP: 1-methyl-4-phenylpyridinium, 6-OHDA: 6-hydroxydopamine, SN: substantia nigra, PD: Parkinson’s disease.

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
