# Peer review of "CX3CL1 Regulation of Gliosis in Neuroinflammatory and Neuroprotective Processes"

_ijms, 2025, doi:10.3390/ijms26030959_

Round 1

Reviewer 1 Report

Comments and Suggestions for Authors

The manuscript (ijms-3409692) reviewed changes in CX3CL1 and its receptor CX3CR1 in neurodegenerative conditions and animal models, in particular in Alzheimer´s and Parkinson´s diseases. It was concluded that alterations of CX3CL1 and CX3CR1 expression and synthesis are key in the progression of neuroinflammation and the associated neuronal loss and are quantifiable indicators of the progression of certain neurological diseases thus new therapeutical strategies may be developed based on the modulation of the CX3CL1/CX3CR1 communication system.

Overall, the topic is interesting despite numerous recent reviews in this field. My main suggestion to the authors is that more mechanistic insights can be provided, given in particular many controversy and conflicting reports in this field, and some advice on future research of CX3CL1-CX3CR1 can be made, rather than simply summarizing the literature. In addition, the fact of the inconsistency literature can be emphasized in the conclusions. Is there a common rule in terms of CX3CL1/CX3CR1 roles among the different neurodegenerative conditions? I also have a couple of more comments for the authors:

1.      How about this study? - Analysis of CX3CR1 haplodeficiency in male and female APPswe/PSEN1dE9 mice along Alzheimer disease progression. Hemonnot-Girard AL, Valverde AJ, Hua J, Delaygue C, Linck N, Maurice T, Rassendren F, Hirbec H. Brain Behav Immun. 2021 Jan;91:404-417

2.      Fernandes et al. (Cells. 2022 Jan 1;11(1):137) reported differential changes in CX3CR1 expression in a mice model.

3.      Kim et al 2022 (Alzheimers Dement (Amst). 2022 Sep 20;14(1):e12354) reported down-regulation of CX3CR1 in AD.

4.      Abstract: it is suggested that CX3CL1 is ‘a specific messenger used by neurons to communicate with microglia’. However, as indicated in the text, CX3CL1 is not really specific to neurons. Indeed, it has been shown that activated astrocytes could have increased expression of CX3CR1 (Wilhelmsson et al. Cereb Cortex. 2017 Jun 1;27(6):3360-3377). In addition, ‘or’ in line 14 should be ‘and’; in line 18, the comma should be removed.

5.      Tables were not cited in the text. Table 1 can also include studies on CX3CR1 changes. References 20 and 22 in Table 1 and reference 52 in Table 2 lack the year of publication.

Author Response

We appreciate the constructive comments provided by the reviewer and have attempted to address them in the revised version (a detailed listing is below). Changes in the text are in red to facilitate their revision.

-Following the reviewer indications, new comments have been included in the conclusions section.

-The three references mentioned by the reviewer have been included in the text and in the tables. We thank the reviewer for this valuable input.

-The abstract has been modified following the reviewer´s corrections, removing the word “specific” and including the indicated modifications in lines 14 and 18.

-Tables have been cited in the text, including CX3CR1 changes in table 1 and the missing publication years.

Reviewer 2 Report

Comments and Suggestions for Authors

The review focuses on the role of C-X3-C motif chemokine ligand 1 (CX3CL1), or fractalkine, in neuroinflammation and neurodegeneration, highlighting its dual function as a soluble and membrane-bound chemokine and its critical role in neuron-microglia communication. CX3CL1 is presented as a key regulator of microglial activation and a promising therapeutic target for neuroinflammatory processes. The authors also highlight the involvement of the CX3CL1-CX3CR1 signalling axis in neurodegenerative diseases, such as Alzheimer’s disease (AD) and Parkinson’s disease (PD, summarizing relevant studies and detailing observed alterations in this pathway. However, the authors also recognize the challenges resulting from inconsistent results due to variability in animal and cellular models and limitations in studies of human samples. Overall, the manuscript offers an in-depth analysis of the current knowledge on CX3CL, its receptor and on their role in neuroinflammation and disease progression.

The manuscript is well structured, and I will recommend the acceptance after minor revisions (listed below).

- To provide a more comprehensive perspective, the authors might consider including further information on how mitochondrial dysfunction, potentially triggered by factors such as copper toxicity (DOI: 10.3390/life11050386), might intersect with neuroinflammatory pathways involving CX3CL1 signalling. Since the copper toxicity disrupts mitochondrial proteins and apoptotic pathways this could contribute to neuronal damage and potentially interacting with CX3CL1-mediated signalling to accelerate disease progression.

- Authors could verify that the font style, size, and text alignment used throughout the manuscript adhere to the journal’s formatting guidelines.

- The authors could possibly add a figure illustrating the interaction between CX3CL1-CX3CR1 signalling and mitochondrial dysfunction in neuroinflammation.

Author Response

We appreciate the constructive comments provided by the reviewer and have attempted to address them in the revised version (a detailed listing is below). Changes in the text are in red to facilitate their revision.

-Following the reviewer indications, the reference mentioned suggested has been included in the text. We thank the reviewer for this valuable input.

-The font style, size and alignment have been adjusted to the journal´s requirements.